# Production of ACE Inhibitory Peptides from Whey Proteins Modified by High Intensity Ultrasound Using Bromelain

**DOI:** 10.3390/foods10092099

**Published:** 2021-09-05

**Authors:** Lucía Abadía-García, Eduardo Castaño-Tostado, Anaberta Cardador-Martínez, Sandra Teresita Martín-del-Campo, Silvia L. Amaya-Llano

**Affiliations:** 1Facultad de Química, Universidad Autónoma de Querétaro, Querétaro 76010, Mexico; lucia.abadia@uaq.mx (L.A.-G.); ecastano@uaq.mx (E.C.-T.); 2Tecnologico de Monterrey, Escuela de Ingeniería y Ciencias, Querétaro 76130, Mexico; mcardador@tec.mx (A.C.-M.); smartinde@tec.mx (S.T.M.-d.-C.)

**Keywords:** antihypertensive peptides, ultrasound pretreatment, whey protein hydrolysate, Fourier-transform infrared spectroscopy by attenuated total reflectance (FTIR-ATR)

## Abstract

High Intensity Ultrasound (HIUS) can induce modification of the protein structure. The combination of enzymatic hydrolysis and ultrasound is an interesting strategy to improve the release of the Angiotensin-Converting Enzyme (ACE) inhibitory peptides. In this study, whey proteins were pretreated with HIUS at two levels of amplitude (30 and 50%) for 10 min, followed by hydrolysis using the vegetable protease bromelain. The hydrolysates obtained were ultrafiltrated and their fractions were submitted to a simulated gastrointestinal digestion. The conformational changes induced by HIUS on whey proteins were analyzed using Fourier-transform infrared spectroscopy by attenuated total reflectance (FTIR-ATR) and intrinsic spectroscopy. It was found that both levels of ultrasound pretreatment significantly decreased the IC_50_ value (50% Inhibitory Concentration) of the hydrolysates in comparison with the control (α *=* 0.05). After this treatment, HIUS-treated fractions were shown as smaller in size and fractions between 1 and 3 kDa displayed the highest ACE inhibition activity. HIUS promoted significant changes in whey protein structure, inducing, unfolding, and aggregation, decreasing the content of α-helix, and increasing β-sheets structures. These findings prove that ultrasound treatment before enzymatic hydrolysis is an innovative and useful strategy that modifies the peptide profile of whey protein hydrolysates and enhances the production of ACE inhibitory peptides.

## 1. Introduction

Whey is the liquid by-product obtained in the cheesemaking process and comprises between 85 and 90% of the initial milk volume. Whey is a rich source of nutrients, particularly whey proteins that account for 20% of the total milk proteins, and it possesses high biological value [1,2]. Whey proteins are a heterogeneous group of soluble proteins that include β-lactoglobulin (β-lg), α-lactalbumin (α-La), Bovine Serum Albumin (BSA), Immunoglobulins (Igs), and Glycomacropeptide (GMP). Whey proteins are recognized for their nutritional value due to the presence of sulfur and branched-chain amino acids and they are commonly used in food products and recognized as safe (GRAS) [3].

Whey proteins are also an interesting source of nutraceutical components such as bioactive peptides (BAPs). BAPs have been defined as specific fragments of a protein that exert different physiological effects on health [4]. BAPs from whey proteins have been studied for their potential to inhibit the Angiotensin-I-Converting Enzyme (ACE; EC 3.4.15.1). This enzyme plays an important role in the Renin–Angiotensin system and, therefore, on hypertension control [5]. ACE hydrolyzes angiotensin I into the potent vasoconstrictor peptide angiotensin II. Therefore, ACE inhibition is key in controlling high blood pressure, and ACE inhibitory peptides could be used as a nutraceutical strategy for hypertension management and prevention. Antihypertensive peptides are of special interest due to the prevalence of hypertension worldwide, which affects about 30% of the adult population. BAPs could represent a healthier and natural alternative to common drugs [6].

The production of whey protein hydrolysates containing BAPs includes a step of enzymatic hydrolysis of whey protein using enzymes from animal, microbial, and vegetable origins, and a purification step carried out with membrane separation techniques, usually including ultrafiltration. The challenge in the production of BAPs is to obtain a high degree of hydrolysis (DH, and produce short-chain peptides with high activity. Selected enzymes with a hydrolysis time of 4–100 h are used during the hydrolysis, while maintaining pH, temperature, enzyme activity [7]. Moreover, whey proteins are complex substrates due to their compact globular structure that makes them resistant to the enzymatic action, obtaining low DH values. The use of vegetable proteases in BAP production is less conventional than animal or microbial proteases. Bromelain (EC 3.4.22.32), a cysteine protease from pineapple, is widely used in food processing due to its GRAS status; it can produce high hydrolysis degree, showing potential for BAP production [8].

Ultrasound is an emerging processing technology used to enhance the efficiency of several food processing systems. High intensity ultrasound (HIUS) is found between 20 and 100 kHz with a power intensity of 10–1000 W/cm^2^, and it can produce high shear and mechanical energy due to the cavitation phenomenon. Acoustic cavitation, the main mechanism involved in the effect of ultrasound waves, is caused by the delivery of a high amount of energy bubbles in a liquid medium [9]. HIUS instrumentation is highly developed compared with other recent technologies and its scale-up facility is a fundamental reason for exploring different applications. Ultrasound treatment before enzymatic hydrolysis has recently been used for the production of bioactive peptides, finding many advantages, such as hydrolysis time reduction, bioactivity increase, and enzyme reduction, compared with traditional enzymatic hydrolysis [10,11,12]. The conformational changes in the structure of proteins have been associated with the application of ultrasound, which weakens interactions and disrupts quaternary and tertiary structures, thus increasing enzymatic hydrolysis [13]. These, results can inform the study of structural changes in proteins and their effects on bioactivity.

The use of food by-products, such as cheese whey, is an emerging topic due to its potential contribution to the sustainability of food processes. Protein hydrolysis assisted by ultrasound technology could be applied to modify the complex protein structure of whey proteins and enhance the enzymatic hydrolysis, releasing short-size BAPs with ACE inhibitory activity. In this study, we aimed to evaluate the conformational modifications of the secondary structure of whey proteins produced by HIUS pre-treatment, by using intrinsic fluorescence and Fourier-transform infrared spectroscopy by attenuated total reflectance (FTIR-ATR) analyses. Additionally, we evaluated the effects of ultrasound on the ACE inhibitory activity of whey protein hydrolysates and their ultrafiltrate fractions produced with the vegetable enzyme bromelain.

## 2. Materials and Methods

### 2.1. Materials

The whey protein used was whey protein isolate (WPI), provided by Hilmar Ingredients (Hilmar, CA, USA), which contained 91.7% protein, 0.5% lactose, and 1.4% fat. Bromelain from pineapple (BROMELAIN 240^®^ EC 3.4.22.32), with an activity of 300 Casein digestion units (CDU), was kindly provided by Enzyme Development Corp. (New York, NY, USA). Angiotensin-converting enzyme (ACE) from rabbit lungs, containing 0.25 units/mg protein (EC 3.4.15.1), hippuryl-histidyl-leucine (HHL), trifluoroacetic acid (TFA), and hippuric acid, were purchased from Sigma-Aldrich (St Louis, MO, USA).

### 2.2. Sample Preparation and Ultrasonic Pretreatment 

WPI powder was suspended in distilled water to obtain solutions at 10, 20, 30, 40, and 50 g/L protein substrate concentration; they were stirred for 30 min and then allowed to stand at 4 °C for complete hydration. The WPI solutions were pretreated using an ultrasound homogenizer (500 W nominal power at 20 kHz) with a 13 mm titanium diameter probe (Cole-Parmer Instrument Co., Vernon Hills, IL, USA) at 25 and 50% of amplitude. During sonication, the solutions were jacketed with chilled water and the temperature did not exceed 25 °C.

The acoustic power intensity, Ia (W/cm^2^), was determined by calorimetry, recording the temperature as a function of time using the equation of Margulis and Margulis [14]:(1)$$Ia=\frac{Pa}{S_{A}}\;\mathrm{where}\;P=m\times Cp\times \left(\frac{dT}{dt}\right)$$
where Pa (W) is the acoustic power, $S_{A}$ is the surface of the ultrasound emitting surface (5.06 cm^2^), m is the mass of the sonicated liquid (g), Cp is the specific heat capacity at a constant pressure (J/Kg °C), and dT/dt is the slope at the origin of the curve.

### 2.3. Enzymatic Hydrolysis of WPI Solutions

After ultrasonic treatment, the temperature of WPI solutions rose and was maintained constant at 55 °C in a jacketed glass reactor, and the pH was adjusted at 7.0 using NaOH 1 M solution. Bromelain enzyme (enzyme/substrate ratio 1:20 *w*/*w*) was added to the solutions and hydrolysis was performed by constant agitation for 1 h. Protein hydrolysis was stopped with enzyme deactivation by heating the mixture for 15 min at 90 °C. Control enzymolysis treatment was performed without ultrasonic pretreatment.

### 2.4. ACE Inhibition of WPI Hydrolysates

ACE inhibitory activity was measured by the Cushman and Cheung method [15] with some modifications. The hydrolysates (0.08 mL) mixed with 0.2 mL of 5 mM HHL (dissolved in a pH 8.3 borate buffer with 0.3 M NaCl) were preincubated for 5 min at 37 °C. The reaction was initiated by adding 0.02 mL of ACE (0.1 units/mL in distilled water) and terminated by the addition of 0.25 mL of 1 N HCl after 30 min of incubation at 37 °C. Next, the mixture was filtered through a 0.22 µm membrane for analysis by Reverse-Phase High-Performance Liquid Chromatography (RP-HPLC). For this analysis, a Agilent ZORBAX 300SB-C18 column (Agilent technologies, Waldbronn, Germany) was used, and the hippuric acid formed in the enzymatic process was detected at 228 nm. The elution flux was 0.8 mL/min with a two solvent system (A) 0.1% TFA in water and (B) 0.1% TFA in acetonitrile. The solvent system was 80% of solvent A and 20% of solvent B during 12 min. The ACE inhibitory activity was expressed as an IC_50_ value, which is defined as the concentration of hydrolysate (µg/mL) necessary to reduce the activity of the enzyme by 50%. The IC_50_ was determined using graphical extrapolation by plotting ACE inhibition as a function of different hydrolysate concentrations.

### 2.5. Hydrolysate Fractionation by Ultrafiltration 

The whey protein hydrolysate with the highest ACE-AI (with and without ultrasonic pretreatment) was selected for sequential ultrafiltration using a crossflow flat-sheet membrane unit, SEPA^®^ CF Membrane Cell System (GE Osmonics, Minnetonka, MN, USA). The membranes used in this study were with molecular weight cut-off (MWCO) of 10 kDa (HFK-131, Koch Membrane Systems, Inc., Wilmington, MA, USA), 5 kDa (HFK-328, Koch), 3.5 kDa (GK, GE Osmonics), and 1 kDa (GE, GE Osmonics). The hydrolysates were ultrafiltrated on sequential mode. First, the complete hydrolysate was filtered using the 10 kDa membrane, the permeate was used as a feed to the second ultrafiltration (UF) step using 5 kDa membrane, and this second permeate was fed to the third UF step using 3.5 kDa membrane and, finally, this permeate was filtered by 1 kDa membrane (Figure 1). Five fractions were collected: F1 (>10 kDa), F2 (5–10 kDa), F3 (3.5–5 kDa), F4 (1–3.5 kDa), and F5 (<1 kDa). The protein content of each fraction was determined using the Kjeldahl analysis, as described in the AOAC Method 991.20 [16], and multiplying by a factor of 6.38.

### 2.6. Simulated Digestion of UF Fractions 

The total hydrolysates (TH) and the five UF fractions were submitted to in vitro digestion using the methodology of Majumder and Wu [17]. The temperature of the samples was adjusted at 37 °C and the pH was adjusted to 2 by adding 1 N HCl. The samples were first digested by pepsin (4% *w*/*w* of protein) for 3 h, then the pH was increased to 7.5, to inactivate the enzyme, by adding 1 N NaOH solution, and then they were subjected to pancreatin digestion (2% *w*/*w* of protein) for another 3 h. The hydrolysis was terminated by raising the temperature to 95 °C and maintaining it for 10 min. The ACE inhibition activity of the hydrolysates and UF fractions before and after simulated digestions were determined, as was described above in Section 2.4

### 2.7. Intrinsic Fluorescence Spectra

Conformational changes of induced WPI solutions were monitored by intrinsic tryptophan fluorescence spectra. WPI solutions of 30 g/L, sonicated at 25 and 50% of amplitude and non-sonicated as native control, were diluted at a final concentration of 0.2 mg/mL of protein. Fluorescence measurements were performed using a Cary Eclipse Fluorescence Spectrophotometer (Agilent Technologies, Santa Clara, CA, USA). The solutions were exposed to an excitation wavelength of 295 nm. The fluorescence emission spectrum was collected in the range of 300 to 450 nm and the fluorescence intensity was expressed as arbitrary units. All measurements were performed at room temperature (~22 °C).

### 2.8. Fourier-Transform Infrared Spectroscopy

The freeze-dried sonicate and non-sonicated WPI solutions were analyzed by Fourier-transform infrared spectroscopy using a Cary 630 FTIR Spectrometer (Agilent Technologies, Santa Clara, CA, USA) equipped with a diamond attenuated total reflectance (ATR). Each spectrum was obtained over 64 scans, with wavelengths ranging from 4000 to 600 cm^−1^ and a resolution of 4 cm^−1^. The protein spectra baseline was corrected, and the water spectrum was subtracted according to Martín del campo et al. [18]. Quantitative analysis of the changes in the secondary structure of WPI samples was determined from amide I region (1700–1600 cm^−1^), the amide I spectra obtained were analyzed by OMNIC software (Version 8, Thermo Nicolet Co., Madison, WI, USA). For the location of the bands for each secondary structures (α-helix, β-turns, and random coils) each spectrum was deconvoluted. A half-bandwidth of 23 cm^−1^ and an enhancement factor of 3 with triangular apodization was employed. A curve-fitting procedure was performed with Gaussian shape, using OriginPro software (Version 9, Origin Lab Corporation, Northampton, MA, USA). The bands from 1600 to 1640 cm^−1^ and 1674 to 1680 cm^−1^ were assigned to β-sheets. The bands from 1671 to 1647 cm^−1^ were assigned to random coil. The bands between 1648 to 1660 cm^−1^ were assigned to α-helix, and the bands in 1663 cm^−1^, 1671 cm^−1^, 1683 cm^−1^, 1683 cm^−1^, 1688 cm^−1^, and 1694 cm^−1^ were assigned to β-turns. The peaks between 1600 cm^−1^ and 1619 cm^−1^ were considered as aromatic side-chains, according to Haque et al. [18].

## 3. Results and Discussion

### 3.1. Effect of Substrate Concentration and Ultrasound Pretreatment on ACE Inhibitory Activity

The ACE inhibitory activity (ACE-IA) of the hydrolysates was determined in vitro and expressed in terms of IC_50_ (µg protein/mL). The values of IC_50_ using different substrate concentrations are presented in Figure 1. The IC_50_ values of the WPI hydrolysate show an increase when the substrate concentration increases. Using ultrasound pretreatment, the IC_50_ value of the hydrolysates decrease in both amplitudes applied compared with the control without sonication (*p* = 0.0032). The most remarkable increase in the ACE-IA was observed in the substrate concentration of 30 g/L and sonication of 50% amplitude ($Ia=2.71\;W/{\mathrm{cm}}^{2}$), where a reduction of 42% in IC_50_ concentration was observed. The effectiveness of ultrasound pretreatment to increase bioactivities on hydrolysates from different protein sources was studied. The bioactivity increase was suggested to be due to structural and conformational modifications induced by cavitation bubbles, which could change the susceptibility of the protein substrate exposing more enzymatic attack sites and lead the release of new ACE inhibitory peptide sequences. 

The hydrolysate bioactivity was affected by the interaction of ultrasound pretreatment and substrate concentration (*p* = 0.026). The results show that the increase in substrate concentration from 10 to 50 g/L of whey proteins, applying an ultrasonic treatment of 25%, shows a decrease in the ACE-IA in the hydrolysates. At the higher level of ultrasound amplitude applied (50%), the ACE inhibition was not clearly dependent on whey protein concentration. This result shows the complexity of modifications made by ultrasound pretreatment. Ultrasonic waves are very complex, and the bioactivity can increase or decrease depending on the conditions and the intensity of the treatment, and the viscosity of the medium could modify how the ultrasound waves travel. Shah et al. [19] proposed that with an increase in viscosity in the medium, the absolute reflection factor decreases exponentially. Changes in the intensity or direction of waves could vary the modifications made by ultrasound pretreatment.

### 3.2. Membrane Fractionation of Ultrasound Pretreated Hydrolysates

Selective ultrafiltration of bromelain WPI hydrolysates was performed by a crossflow unit. A substrate concentration of 30 g/L and an ultrasound pretreatment of 50% amplitude were selected for the highest ACE inhibition (IC_50_ 45.07 ± 1.35 µg protein/mL). A control treatment without bromelain was evaluated in order to analyze the effect of HIUS alone on the release of ACE inhibitory peptides. There was negligible ACE inhibitory activity found (IC_50_ 8.20 ± mg protein/mL), indicating that the ultrasonic conditions applied could not cause the peptide bonds breakage in whey proteins. Protein quantification and ACE-IA of each UF fraction were compared with the hydrolysate, with the same substrate concentration but without sonication, with the aim to identify the impact of ultrasound pretreatment on the molecular weight distribution and bioactivity of the hydrolysate. Figure 2 shows the protein balance of each fraction. Protein content in F1 and F2 decreases when ultrasound pretreatment is applied, which means that a higher proteolytic activity was achieved by the enzyme. On the other hand, the fractions of shorter molecular weight (F3, F4, and F5) increase their concentration with the pretreatment. These changes on peptide distribution represent a higher concentration of peptides with a molecular weight below 5 kDa, which could improve their bioactivity. Several studies observed that shorter peptide sequences, generally between 3 and 12 amino acids, exhibited stronger ACE inhibition, mainly due to a higher interaction with the active site of the enzyme [20,21].

Figure 3 presents the ACE-IA (IC_50_) of each associated membrane fraction obtained. Membrane fractionation affected the ACE inhibitory activity: five associated membrane fractions with and without ultrasound pretreatment displayed lower ACE IC_50_ compared with the whole hydrolysate. F3 (5 kDa) and F4 (3 kDa) showed the lowest IC_50_ values, according to previous reports of refining hydrolysates by ultrafiltration. The ultrasound treatment before enzymatic hydrolysis increased the ACE-IA, and the IC_50_ of F2 (10 kDa), F3 (5 kDa ), and F4 (3 kDa) had a reduction of 32%, 43%, and 31%, respectively, compared to the non-sonicated control fractions. However, F1(<1 kDa) and F5 (>10 kDa) did not show statistical differences (α = 0.05). Uluko et al. [20] reported similar results; however, they did not compare the fractions without ultrasound pretreatment. This study showed the potential of ultrasound pretreatment to change the peptide profile and improve the ACE-IA of hydrolysate fractions to produce higher amounts of peptides, thereby increasing their bioactivity.

### 3.3. Stability of ACE Inhibitory Activity of Associated Membrane Fractions Peptides after Simulated Gastrointestinal Digestion

The stability of ACE inhibitory peptides exposed to gastrointestinal enzymes was evaluated via a two-stage hydrolysis process which was intended to simulate in vitro the conditions prevailing during human digestion. The ACE-IA of the whole WPI hydrolysates (WPI-H) and their associated membrane UF fractions with and without ultrasound pretreatment, before and after simulated digestion, is shown in Table 1. The IC_50_ value of the whole WPI hydrolysates decreases after digestion without ultrasound pretreatment, while no significant change was observed on hydrolysates with US pretreatment (α = 0.05). The main differences observed in the whole WPI-H bioactivity due may be to the different profiles of undigested proteins and large peptides contained, that are the target for proteolytic enzymes during the digestion assay.

Associated membrane UF fractions above of 10 kDa, F1, and F2, present a decrease in IC_50_ value after digestion. These results are according to Wang et al. [22], who suppose that gastrointestinal digestion may release and form active fragments, from inactive or fewer active precursors. On the other hand, associated membrane fractions between 1 and 5 kDa (F3 and F4), showed an increase in IC_50_ value; these fractions showed the same pattern with and without HIUS pretreatment. The decrease in the ACE inhibitor effect is probably due to the enzyme attack on active peptic fragments where the C-terminal amino acid residues were hydrolyzed, and the resulting peptides are less compatible with the active site of ACE. It is important to notice that residual ACE-IA is remarkable after simulated digestion, and the ACE-IA of F1 and F2 pretreated with HIUS are higher than non-sonicated ones. The F5, mainly amino acids and short peptides, remained stable after in vitro digestion. This effect may be due to peptides as these are less susceptible to the attack of proteolytic enzymes.

### 3.4. Analysis of Fluorescence Spectra

Changes in the maximum wavelength and fluorescence intensity were evaluated to study the structural changes of whey proteins induced by ultrasound treatment. Ultrasound pretreatment did not alter the shape of the WPI fluorescence spectrum, and typical fluorescence emission spectra with a maximum at 332 nm were observed. This fluorescence spectrum is characteristic of tryptophan residues of the two main whey proteins, α-La and β-Lg which have two and four tryptophan residues, respectively [23]. This indicated no changes in the microenvironment of tryptophan residues, and these results are in agreement with Gao et al. [24]. Figure 4 shows that the fluorescence intensity decreases with the increase in amplitude of ultrasound pretreatment. This could be due to the quenching of tryptophan fluorescence indicating structural changes, aggregation, or oxidation exposed to ultrasound. Tryptophan fluorescence can be quenched by water molecules, and amino acids positively charged, such as cysteine, histidine, or tyrosine [25].

### 3.5. FTIR-ATR Spectral Analysis

The amide I region (1600–1700 cm^−1^) of the FTIR spectra of WPI solutions with and without ultrasonic pretreatment of 25 and 50% amplitude were used to evaluate changes in the secondary structure of proteins. The amide region-I corresponds mainly to the stretching vibration of C=O and C-N bonds [26]. As is shown in Figure 5, there is an alteration on the secondary structure of WPI due to the ultrasound treatment. There is an increase in band intensity within this region. The secondary structural changes of proteins were not completely visible from the FTIR spectra, hence the amide I band was deconvoluted, the peaks were located and assigned to a secondary structure, and the percentage of secondary structures were calculated using a Gaussian method (Table 2). The content of secondary structures of WPI without HIU treatment was consistent with other FTIR studies [27,28]. The modification of the secondary structures for HIUS treatment was statistically significant, in general terms, and found a decrease in α-helix and disordered structures (random coil), and an increase in β-sheets. The changes in the percentages of β-turns were dependent on the sonic amplitude applied. While at 25% amplitude there were no changes (α = 0.05), at 50% of amplitude there was a significant increase (*p* = 0.012). Changes in α-helix composition are attributed to the cavitation phenomenon associated with the ultrasonic treatment, and this could show a partial unfolding of α-La. The increase in β-sheet structures is related to changes in the intra or intermolecular hydrogen bonds exposing the hydrophobic amino acids [29]. These changes are promoted by the acoustic energy applied sufficiently to reduce the number of hydrogens bonds and electrostatic interactions. Then, this new β-sheets result is from a subsequent crosslinking of proteins, aggregation, and network formation [30,31,32].

## 4. Conclusions

The HIUS treatment on WPI solutions depends on the acoustic parameters used (amplitude and time), and the whey protein concentration. An ultrasound treatment before enzymatic hydrolysis with bromelain modified the pattern of the WPI hydrolysates. The peptides released possessed higher ACE inhibitory activity compared to their controls with no ultrasound-treated hydrolysates. The most remarkable changes were detected in the peptide fractions between 1 and 5 kDa. This ACE inhibitory activity was stable after simulated gastric and intestinal digestion. However, further studies of the molecular mass peptide profile will be necessary to obtain a broad understanding of the changes. 

Results of the study of conformational changes showed that the ultrasound induces modifications on whey protein producing changes of the secondary structures. The FTIR and fluorescence spectrum showed that HIUS affects the whey protein conformation and generates new rearrangements between proteins that unfold, and finally produces aggregates. This study showed that ultrasound treatment can improve the enzymatic process with bromelain to obtain ACE inhibitory peptides from whey proteins that are suitable for use as food ingredients. This could enhance the feasibility of bioactive peptide production adding value to whey proteins.

## Figures and Tables

**Figure 1 foods-10-02099-f001:**
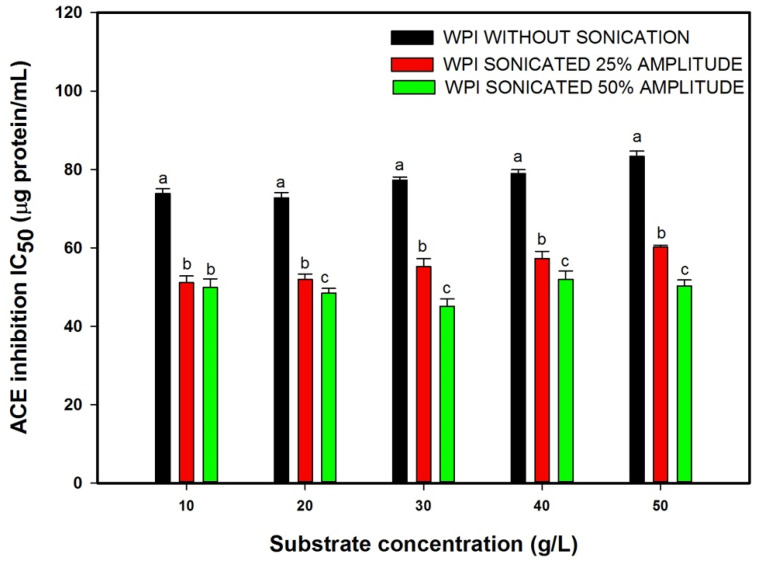
Effect of substrate concentration and ultrasonic treatment on the angiotensin-converting enzyme (ACE) inhibitory activity of whey protein isolate (WPI) hydrolysates. Values are means with their standard deviations depicted by vertical bars. Different letters indicate significant differences (α = 0.05, Dunnett’s test) within the same group.

**Figure 2 foods-10-02099-f002:**
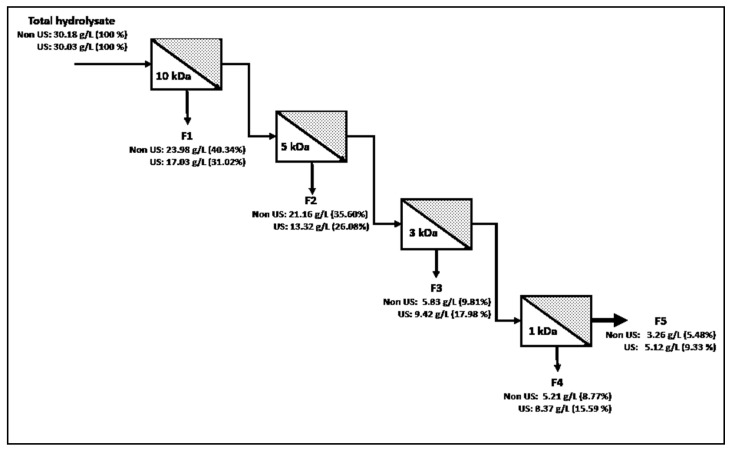
Flow chart of membrane fractionation of WPI hydrolysate and the protein concentration of each associated membrane fraction.

**Figure 3 foods-10-02099-f003:**
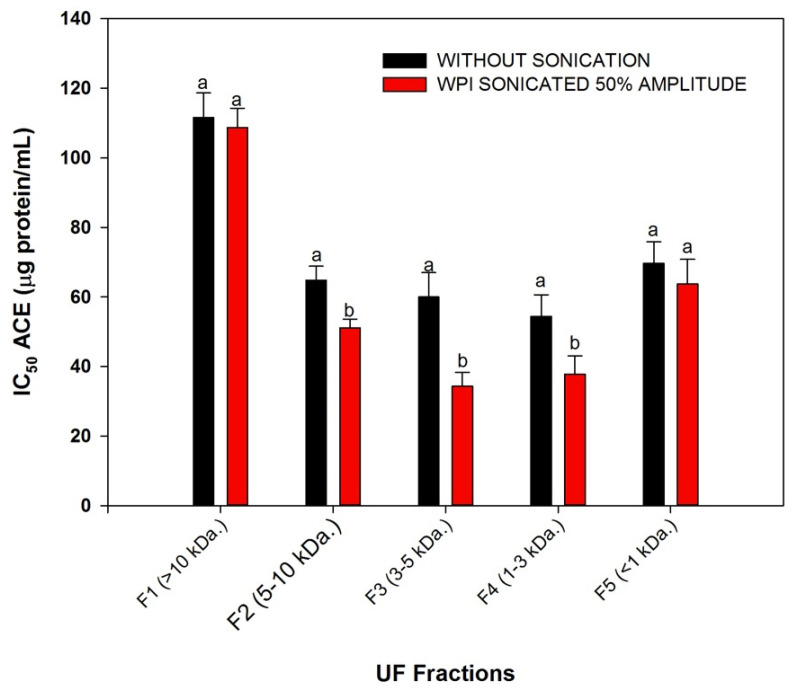
ACE inhibitory activity of associated membrane fractions with and without ultrasound treatment. Values are means with their standard deviations depicted by vertical bars. Different letters indicate significant differences between the sonicated treatment and its control without sonication (Dunnett’s test, α = 0.05).

**Figure 4 foods-10-02099-f004:**
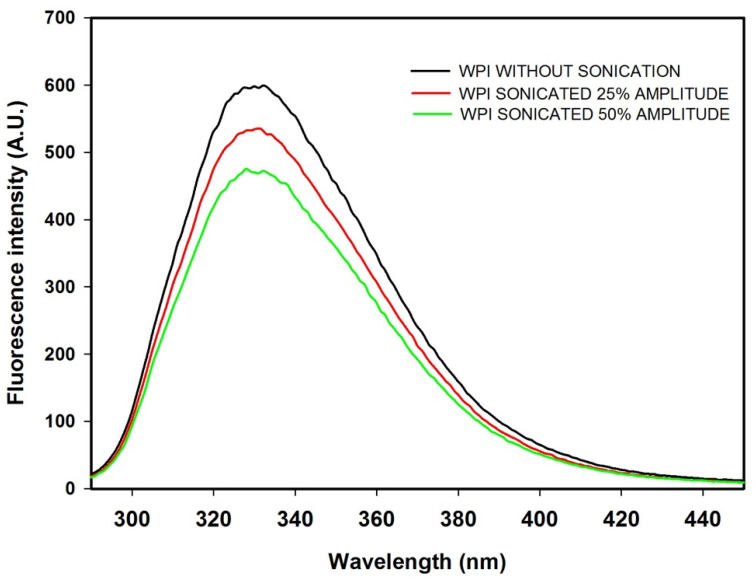
Effect of ultrasound treatment (25 and 50% of amplitude) on intrinsic fluorescence spectrum.

**Figure 5 foods-10-02099-f005:**
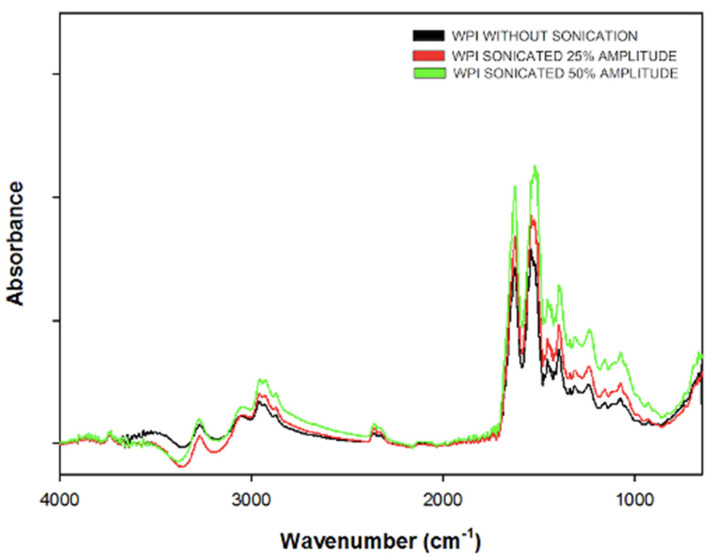
Effect of ultrasound treatment (25 and 50% of amplitude) on Fourier-transform infrared spectroscopy by attenuated total reflectance (FTIR-ATR) spectra of WPI samples.

**Table 1 foods-10-02099-t001:** ACE inhibitory activity of whey protein isolate hydrolysates (WPI-H) and their associated membrane fractions with and without High intensity ultrasound (HIUS) treatment, before and after simulated in vitro digestion conditions.

	ACE Inhibitory Activity (IC_50_ µg Protein/mL)			
	Without HIUS Pretreatment		With HIUS Pretreatment	
Samples	Before Simulated Gastrointestinal Digestion	After Simulated Gastrointestinal Digestion	Before Simulated Gastrointestinal Digestion	After Simulated Gastrointestinal Digestion
**WPI Hydrolysate (WPI-H)**	87.29 ± 0.08 ^a^	68.21± 2.15 ^b^	45.07 ± 4.90 ^b^	41.08 ± 5.23 ^b^
**F1 (>10 kDa)**	111.60 ± 7.06 ^a^	74.56 ± 4.58 ^b^	108.65 ± 5.58 ^a^	63.00 ± 10.12 ^b^
**F2 (5–10 kDa)**	64.80 ± 4.12 ^a^	53.15 ± 5.68 ^b^	51.10 ± 2.51 ^a^	45.15 ± 2.30 ^b^
**F3 (3–5 kDa)**	60.03 ± 7.09 ^a^	78.45 ± 4.12 ^b^	34.30 ± 4.02 ^a^	72.05 ± 6.12 ^b^
**F4 (1–3 kDa)**	54.40 ± 6.24 ^a^	62.14 ± 3.52 ^b^	37.80 ± 5.20 ^a^	58.07 ± 5.10 ^b^
**F5 (˂1 kDa)**	69.60 ± 6.32 ^a^	69.15 ± 4.57 ^a^	63.70 ± 7.10 ^a^	61.14 ± 6.51 ^a^

Values represent the means ± standard error (n = 3); different superscript letters in the same line indicate significant differences (Tukey HSD test, α = 0.05).

**Table 2 foods-10-02099-t002:** Secondary structure analysis of WPI samples in amide I region.

	Secondary Structure Composition (%)				
Samples	R^2^	α-Helix	β-Sheet	β-Turn	Random Coil
Non-sonicated WPI	0.97	12.01 ± 0.24 ^a^	31.99 ± 0.17 ^a^	34.44 ± 0.28 ^a^	21.58 ± 0.19 ^a^
Ultrasound-treated WPI 25% amplitude	0.98	9.81 ± 0.19 ^b^	39.56 ± 0.21 ^b^	34.97 ± 0.23 ^a^	15.66 ± 0.18 ^b^
Ultrasound-treated WPI 50% amplitude	0.97	9.59 ± 0.22 ^b^	36.43 ± 0.19 ^b^	37.10 ± 0.32 ^b^	16.89 ± 0.26 ^b^

Values represent the means ± standard error (n = 3); different superscript letters in the same column show a significant difference (Dunnett’s test, α = 0.05).

## Data Availability

Data available upon request.

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
