# Peer review of "Production of ACE Inhibitory Peptides from Whey Proteins Modified by High Intensity Ultrasound Using Bromelain"

_foods, 2021, doi:10.3390/foods10092099_

Round 1

Reviewer 1 Report

In general, the article is interesting, but still needs to be refined. Some fragments are poorly polished in terms of style and linguistics. I have included detailed comments in the attached pdf file.

Author Response

Dear reviewer 1, 

The manuscript Ms. Ref. No.: foods-1286379; Title: Production of ACE inhibitory peptides from whey proteins modified by High Intensity Ultrasound using bromelain has been revised considering all the comments, corrections and suggestions made. Please see the attachment.

We wish to thank  for your valuable comments to avoid misunderstandings and improve the quality of the manuscript, all the comments are addressed as well as the actions taken to improve our contribution

Reviewer 2 Report

This work, despite being extensively studied, is the combination of ultrasonic treatment and enzymatic hydrolysis is interesting  seems to be well carried out and the manuscript is well written. However, I suggest including a control without enzymatic treatment, in order to compare the effects. If the possibility exists, it would be interesting to include an analysis at MS/MS or HPLC level in order to understand the peptide profile of each of the fractions studied.

Author Response

Reviewer 2,

The manuscript Ms. Ref. No.: foods-1286379; Title: Production of ACE inhibitory peptides from whey proteins modified by High Intensity Ultrasound using bromelain has been revised considering all the comments, corrections and suggestions made by the reviewers. Please see the attachment.

We wish to thank the reviewers for their valuable comments to avoid misunderstandings and improve the quality of the manuscript, all the reviewer’s comments are addressed as well as the actions taken to improve our contribution. We thank you for your time and interest in our research.

Best regards

Round 2

Reviewer 1 Report

In present form article can be accept to publication in Foods